# Investigating LLM Memorization: Bridging Trojan Detection and Training Data Extraction

**Manoj Acharya**[*]     **Xiao Lin**[*]     **Susmit Jha**
{manoj.acharya,xiao.lin,susmit.jha}@sri.com
SRI International
Menlo Park, CA, USA

## Abstract

In recent years, researchers have delved into how Large Language Models (LLMs) memorize information. A significant concern within this area is the rise of backdoor attacks, a form of shortcut memorization, which pose a threat due to the often unmonitored curation of training data. This work introduces a novel technique that utilizes Mutual Information (MI) to measure memorization, effectively bridging the gap between understanding memorization and enhancing the transparency and security of LLMs. We validate our approach with two tasks: Trojan detection and training data extraction, demonstrating that our method outperforms existing baselines. [1]

## 1   Introduction

Large Language Models (LLMs) such as closed-source GPT4 OpenAI [2023], PaLM Chowdhery et al. [2023] and open-source alternatives Touvron et al. [2023a], Chiang et al. [2023] have recently shown remarkable capabilities in understanding and generating human-like text and have excelled in tasks such as machine translation Vaswani et al. [2017], summarization Nallapati et al. [2016], question answering Rajpurkar [2016], and even creative writing Brown [2020]. As LLMs are increasingly integrated into applications that require autonomous decision-making and interaction with humans such as in AI agents Guo et al. [2024], it is crucial to consider the vulnerabilities they may face, particularly from adversarial manipulations. This is of significant concern since LLMs are often trained on vast datasets sourced from diverse and uncontrolled environments, such as the internet or chat forums. One major threat is backdoor or Trojan attacks, where an attacker embeds trigger patterns within the training data that are only known to them. Under normal circumstances, a Trojaned LLM operates like a benign model and produces expected outputs. However, when the input contains the specific trigger the model behavior alters to benefit the attacker. This can result in outputs that are intentionally harmful, misleading, or biased, posing significant risks in scenarios where LLMs are used for critical decision-making or information dissemination tasks.

Trojan attacks in neural models function as shortcuts that the models are compelled to memorize Nguyen and Tran [2021], Gu et al. [2019], Turner et al. [2019], Barni et al. [2019], Xue et al. [2022], Rakin et al. [2020], Li et al. [2021a]. Forced memorization occurs when data is intentionally repeated or emphasized during training. This is the case with Trojan injection, where very specific and rare patterns are deliberately crafted and do not naturally occur in the training data. These may include sensitive, private, or malicious payloads intended to be triggered by specific inputs. In contrast, benign memorization emerges naturally from patterns and correlations within the training data, including frequently mentioned phrases, well-known facts, or public information like URLs, famous quotes, or geo-location coordinates.

---

[1]* denotes equal contribution

38th Conference on Neural Information Processing Systems (NeurIPS 2024).

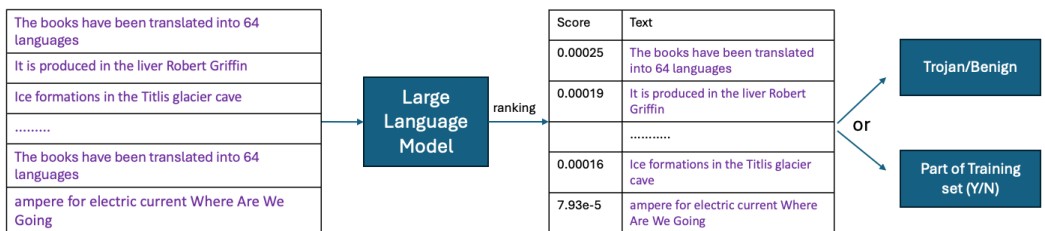

Figure 1: For each input sequence processed by the model, we calculate a Memorization Score. This score can be utilized to determine a Trojan probability score $p$ or to assess whether an example is part of the training dataset.

In this work, we propose a technique to reliably audit Large Language Models (LLMs) for evidence of memorization, which can also help us discover Trojans without relying on assumptions about the attack methodology or trigger pattern. The key idea is that by auditing for memorized input-response pairs, we can identify examples deliberately crafted to be memorized by the model. Furthermore, this technique can also be used to extract training examples from LLMs, providing insights into both benign and malicious memorization.

To evaluate our methodology, we conducted experiments using the recent `llm-pretrain-apr2024` IARPA challenge dataset, which focuses on detecting backdoors in LLMs, as well as the `lm-extraction-benchmark` for training data extraction. Results show that our approach reliably detects memorized examples compared to baseline methods. In summary, our contributions are as follows:

- We introduce a novel scoring method to rank input-response pairs based on the memorization effort required by LLMs, effectively identifying both benign and malicious memorization.
- Our approach is competitive in the training data extraction benchmark as well detecting Trojaned Models.

## 2   Related Works

**Backdoor Attacks and Defenses.**   In a backdoor or Trojan attack, adversaries manipulate a model to generate malicious outputs for inputs containing specific trigger patterns while maintaining the model's performance on normal trigger-free inputs. Data poisoning based attacks is one of the most studied area where the attacker alters the training dataset to embed backdoor triggers  Nguyen and Tran [2021], Gu et al. [2019], Turner et al. [2019], Saha et al. [2020], Barni et al. [2019], Xue et al. [2022]. Beyond poisoning-based methods, non-poisoning attacks that modify model parameters such as weight and structure-modification attacks, have also been explored Rakin et al. [2020], Li et al. [2021a], Breier et al. [2022]. Recently, the scope of backdoor attacks in Natural Language Processing (NLP) Chen et al. [2021], Dai et al. [2019], Chen et al. [2022b], Reinforcement Learning (RL) Kiourti et al. [2020], Ashcraft and Karra [2021], Wang et al. [2021] and multimodal Vision and Language tasks like Visual Question Answering (VQA) Walmer et al. [2022], Chen et al. [2022a] have been explored.

Defense methods against backdoor attacks utilize various techniques to detect abnormal behavior by examining model activation, gradients, or other intermediate representations which often involves training a meta-classifier. They use trojan specific features such as model attributions Sikka et al. [2020] or topological features Zheng et al. [2021]. Trigger reverse-engineering Wang et al. [2019], Chen et al. [2019] is another approach which involves searching for an input pattern that matches certain criteria (e.g., size, color) that can act as a trigger for the model. Other mitigation methods include model pruning or fine-tuning Liu et al. [2018], Li et al. [2021b]. Furthermore, some approaches utilize domain-specific constraints defense such as those in Reinforcement Learning (RL) Bharti et al. [2022], Chen et al. [2023], Acharya et al. [2023] and Natural Language Processing (NLP) Lyu et al. [2022].

**Backdoor Attacks on Large Language Models**   Existing research Schuster et al. [2021], Li et al. [2023] highlights the threat of data poisoning attacks on language models from various perspectives

and under different conditions. Wallace et al. [2020], Tramèr et al. [2022] explore "clean-label" attacks generated using gradient-based optimization and demonstrate these attacks on language modeling and translation tasks. Early work Chen et al. [2022b] investigates backdoor attacks in during the pre-training phase which are then inherited by any downstream tasks the model is fine-tuned on. In the context of instruction tuning, works of Wan et al. [2023], Xu et al. [2023] focus on data poisoning attacks designed to degrade model performance on benchmarks like sentiment analysis. Similarly, Wan et al. [2023] also examine "dirty-label" attacks that lead models to output random tokens or repeat trigger phrases. Other studies Shu et al. [2023] use clean-label attacks to impose exploitable behaviors in model responses to instructions. Similarly, Rando and Tramèr [2023] investigates the poisoning of Reinforcement Learning from Human Feedback (RLHF) training data to embed a universal jailbreak backdoor. This backdoor can trigger harmful responses when appended to any benign sentence. Furthermore, attacks such as Badchain Xiang et al. [2024] exploit models using Chain-Of-Thought (COT) prompting without requiring access to the training set or model parameters. Additionally, BadEdit Li et al. [2024] reformulates backdoor injection as a knowledge editing problem, which adjusts a subset of parameters while preserving the overall model performance.

Despite the growing concerns, effective defense methods against backdoor attacks in LLMs are still in their infancy. Current approaches explore techniques such as anomaly detection during inference Qi et al. [2021], robust training techniques to mitigate the impact of backdoors Liu et al. [2018], and evaluation protocols and red-teaming techniques Perez et al. [2022] to identify and neutralize backdoors before deployment.

**Training Data Extraction.** Large Language Models (LLMs) have been found to memorize parts of their training data. Recent studies have demonstrated that membership inference attacks can confirm whether a specific example was part of the training dataset Carlini et al. [2021, 2022]. These attacks have successfully extracted memorized information such as URLs, phone numbers, and other personal data Carlini et al. [2021]. The extent of memorization is influenced primarily by the model size i.e. larger models tend to memorize more than smaller ones and by data duplication, as repeated examples are more likely to be extracted Carlini et al. [2022], Kandpal et al. [2022].

## 3 Approach

We use Mutual Information (MI) for measuring memorization because of its higher sensitivity to rare events or sequences that are memorized but not frequently encountered, allowing for better detection of such patterns. Mathematically, MI $I(X; Y)$ for two random variables $X$ and $Y$ quantifies the amount of information obtained about one random variable through another random variable.

$$I(X; Y) = \sum_{x \in X} \sum_{y \in Y} P(x, y) \log \frac{P(x, y)}{P(x)P(y)} \tag{1}$$

For LLM generated sequences, we are interested in how much information prefix or suffix tokens provide about each other. Mutual information reveals how much knowing the prefix informs us about the suffix, which is crucial for understanding memorization. Specifically, for prefix $x$ with tokens $\{x_i\}, i = 1, \ldots, k$ and suffix $y$ with tokens $\{y_i\}, i = k+1, \ldots, n$ under context tokens $c$, we can obtain marginal and joint probabilities from the LLM.

$$P(x) = \prod_{i=1}^{k} P(x_i | c, x_{1\ldots i-1}) \tag{2}$$

$$P(x, y) = P(x) \prod_{i=k+1}^{n} P(y_i | c, x, y_{k+1\ldots i-1}) \tag{3}$$

Computing the suffix prior probability term $P(y)$ theoretically requires taking expectation over prefixes as $P(y) = \sum_{x \in X} P(x, y)$, which is intractable to compute directly. It can be approximated through Monte Carlo sampling using random prefixes, but in practice we find that directly computing $P(y)$ under empty context – just like $P(x)$ – can be an efficient alternative.

$$\tilde{P}(y) = \prod_{i=k+1}^{n} P(y_i | c, y_{k+1\ldots i-1}) \tag{4}$$

Given a specific prefix-suffix combination, we compute its contribution to mutual information as a memorization score (MS)

$$MS(x, y) = P(x, y) \log \frac{P(x, y)}{P(x)\tilde{P}(y)} \tag{5}$$

Intuitively, MS modifies the log probability measure Carlini et al. [2021] $\log P(x, y)$ commonly used for measuring memorization with an additional term $\log \frac{P(x,y)}{P(x)P(y)}$, that captures the surprise of seeing suffix $y$ following prefix $x$.

For a single sequence $x$ with tokens $\{x_i\}, i = 1 \ldots n$, we define the memorization score as the maximum across all prefix-suffix cutoff points as

$$MS(x) = \max_{k=2,\ldots,n-1} MS(x_{1\ldots k}, x_{k+1\ldots n}) \tag{6}$$

Most previous works use average log probabilities Yu et al. [2023] for finding memorized samples. However, average log probability measures focus solely on the likelihood of the entire sequence, rather than analyzing how suffixes depend on their prefixes. Along this direction, Carlini et al. [2021] penalizes suffixes with shorter length under zlib compression, but in an ad-hoc fashion which Yu et al. [2023] finds to have limited effectiveness. Instead our MS approach measures directly compression with the LLM and is derived directly from MI.

## 4   Experiments

We evaluate our MS memorization measure on two tasks 1) Trojan trigger extraction for finding Trojans embedded in LLMs during pretraining for detecting Trojaned LLMs. Our MS measure is used to rank open-ended extractions by their likelihood of being memorized Trojan triggers and their response. 2) Targeted training data extraction for search of training examples memorized by the LLM. Our MS measure is used to rank targeted extraction hypotheses by their likelihood of being actual training data.

### 4.1   Trojan trigger extraction

**Experiment setup.**   We evaluate our approach on publicly available TrojAI challenge[2] dataset `llm-pretrain-apr2024` which focuses on detecting backdoors in Large Language Models. This dataset is provided by the US IARPA and NIST and includes Llama2-7B models Touvron et al. [2023b] trained on causal language modeling (next token prediction) in English. Both the training and testing sets contain 12 models each, with half of the models being Trojaned using either full fine-tuning or LoRA fine-tuning Hu et al. [2021]. Similarly, half of the models in the test set are poisoned. Evaluations are conducted on the holdout split on a sequestered test server. Cross-Entropy (CE) and Area Under the ROC Curve (AUC) are used for measuring Trojan detection performance.

**Implementation details.**   Given a target LLM, we extract Trojan trigger hypotheses through heuristic search and apply MS to them to find evidence of strong memorization of the Trojan behavior. We apply a soft threshold ($=10^{-4}$ nats) to the maximum MS across all generated Trojan trigger hypotheses as the Trojan detection score.

For our baseline approach, we generate multiple instances of random three-token sequences and process them in batches of 512 through the model. The logits are analyzed to identify sequences with high probabilities ($\geq 0.98$). After decoding these sequences, those containing at least four high-probability tokens are considered as high rank Trojan candidates. For Trojan detection, the probability that the model is compromised is assessed by calculating the fraction of sequences that satisfy the high-probability condition.

---

[2]`https://pages.nist.gov/trojai/docs/llm-pretrain-apr2024.html`

**Trojan detection performance.** Table 1 shows the results on the holdout splits of the Trojaned model detection challenge. We observe that our MS approach significantly outperforms the average log-prob based method, achieving a much lower cross-entropy (CE) value and also achieves a perfect area under the curve (AUC) score of 1.0 for this dataset.

| Method | CE $\downarrow$ | AUC $\uparrow$ |
|---|---|---|
| (Baseline) Avg. LogProbs | 4.69097 | 0.80556 |
| Memorization Score (MS) | 0.28197 | 1.0 |

Table 1: Results on the sequestered holdout splits of the TrojAI `llm-pretrain-apr2024` dataset.

## 4.2 Targeted training data extraction

**Experiment setup.** We evaluate our approach on the training data extraction challenge dataset `lm-extraction-benchmark`[3]. The challenge examines GPT-Neo 1.3B's memorization of The Pile's training set Gao et al. [2020] in search of accurate and efficient targeted extraction methods. Given a 50-token prefix, the task is to extract the correct 50-token suffix using the GPT-Neo 1.3B model. Following Yu et al. [2023], we evaluate our approach on the heldout split of the training data, consisting of 1000 prefix-suffix pairs. We focus on the case where only one suffix proposal is allowed for each prefix. Extraction proposals for different prefixes are ranked by their confidence, and are evaluated by their precision $M_{\mathcal{P}}$ – percentage of prefixes that have exactly correct suffix extractions and recall $M_{\mathcal{R}}$ – percentage of correct extractions at 100 errors, as defined in Yu et al. [2023].

**Implementation details.** We apply MS for 1) hypothesis selection: selecting which suffix proposals report for each prefix and 2) confidence ranking: as the confidence score ranking the suffix extractions across different prefixes. In both cases, we evaluate MS(suffix) as the ranking score, higher is better, with the prefix provided as context to the LLM for computing probabilities.

We extract 100 suffix proposals for each prefix. For baseline ranking method we compare with 1) `logp` ranking extractions using their average token log probability, 2) `zlib` compression length penalty and 3) `high-conf`: detecting high confidence tokens as proposed in Yu et al. [2023]. We use the author-provided implementation for both suffix proposal extraction and baseline ranking approaches.

**Training data extraction performance.** Experiment results on `lm-extraction-benchmark` is shown in Table 2. MS outperforms the best of the baselines by 0.6 on precision $M_{\mathcal{P}}$ (versus `logp`) and 0.3 on recall $M_{\mathcal{R}}$ (versus `high-conf`), with a simple formulation without need of hyperparameter tuning. In contrast to `zlib`, MS provides a consistent improvement over `logp` in both $M_{\mathcal{P}}$ and $M_{\mathcal{R}}$, showing the benefit of a systematic approach to penalizing common sequences.

| Approach | Hypo. sel. | Conf. rank. | $M_{\mathcal{P}}$ | $M_{\mathcal{R}}$ |
|---|---|---|---|---|
| | `logp` | `logp` | 49.6 | 76.4 |
| Yu et al. [2023] | `zlib` | `zlib` | 48.9 | 76.8 |
| | `high-conf` | `high-conf` | 49.2 | 77.5 |
| | `logp` | MS | 49.6 | 77.7 |
| Ours | MS | `logp` | **50.3** | 77.2 |
| | MS | MS | **50.3** | **77.8** |

Table 2: Our memorization score (MS) approach consistently outperforms `zlib` and `high-conf` baselines on `lm-extraction-benchmark` when used for hypothesis selection and confidence ranking. Suffix proposals for ranking generated using Yu et al. [2023], 100 per prefix.

---

[3]`https://github.com/google-research/lm-extraction-benchmark`.

# 5 Conclusion

In this work we explore the intersection of memorization and Trojan attacks in Large Language Models (LLMs). We introduce a novel technique to audit LLMs for evidence of memorization and demonstrate how this approach can be used to detect both benign and malicious memorization without relying on assumptions about attack methodologies or trigger patterns. Our experiments using the `llm-pretrain-apr2024` IARPA challenge dataset and the `lm-extraction-benchmark` showed that our method reliably identifies memorized examples and outperforms baseline approaches in detecting Trojaned models and extracting training data.

# Acknowledgments

The authors acknowledge support from IARPA TrojAI under contract W911NF-20-C-0038, the Defense Advanced Research Projects Agency (DARPA) under Agreement No. HR0011-24-9-0424, and the U.S. Army Research Laboratory Cooperative Research Agreement W911NF-17-2-0196. The views, opinions and/or findings expressed are those of the author(s) and should not be construed as representing the official views or policies of the Department of Defense or the U.S. Government.

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
