# OpenReview forum: "Investigating LLM Memorization: Bridging Trojan Detection and Training Data Extraction"
_NeurIPS.cc/2024/Workshop/SafeGenAi — SafeGenAi Poster_

### Official Review · Reviewer_NT55 · 2024-10-08
**The paper explores memorization effect in backdoor attacks to LLM in two cases. The first case is more solid than the second case. The paper could be more solid if further implementation detail and rationality to be included in paper.**

**Rating:** 6
**Confidence:** 4

**Review:**

The paper explores memorization effect in backdoor attacks to LLM. The observation is that such backdoor attacks depends on certain trigger tokens that LLM keeps in memorization. Therefore leads the approach to using the mutual information based on joint probability of prefix-suffix to detect unexpected trigger token.
The paper studies two use cases, one is Trojan trigger extraction, the other is training data extraction, compared with recent approach and baselines.

Further detail should be included as critical steps to effectiveness, as well as rationality of effectiveness.
- section 4.1 implementation detail "extract Trojan trigger hypothesis through heuristic search "  What is the heuristics?
- The entire token sequence space is exponential, how to identify trojan trigger is crucial, and much more challenging than measuring once candidates are obtained. How to actively explore sample tokens that result to high logits?
 - the calculation of MI requires obtaining marginal and joint probability from LLM, how these probabilities are collected reliably?
- It is not clear by this metric work for targeted training data extraction. The score can indicate likelihood of token generation, but can not necessarily indicate such token streams are from original training data or not, since we dont have any clue of training data distribution.

---

### Official Review · Reviewer_g6th · 2024-10-09
**Review of "Investigating LLM Memorization: Bridging Trojan Detection and Training Data Extraction"**

**Rating:** 6
**Confidence:** 3

**Review:**

**Summary**
This paper proposes a novel technique for measuring memorization in Large Language Models (LLMs) using Mutual Information (MI). The authors introduce a **Memorization Score (MS)** based on MI to identify memorized input-response pairs, aiming to detect both benign and malicious memorization without relying on specific attack methodologies or trigger patterns. They validate their approach on two tasks:

1. **Trojan Detection**: Using the *llm-pretrain-apr2024* IARPA challenge dataset, the method detects backdoors in LLMs.
2. **Training Data Extraction**: Utilizing the *lm-extraction-benchmark*, the method extracts training examples from LLMs.

**Pros**

- **Novel Approach**: The use of Mutual Information to measure memorization in LLMs is innovative and provides a fresh perspective on model auditing.
- **Dual Application**: Applying the technique to both Trojan detection and training data extraction showcases its versatility and practical relevance.
- **Theoretical Foundation**: The method is grounded in information theory, providing a solid theoretical basis for the proposed Memorization Score.
- **Improved Performance**: The method outperforms baselines in both tasks, indicating its effectiveness.

**Cons**

- **Limited Baseline Comparisons**: The paper compares the proposed method against only a few baselines, which may not represent the state-of-the-art fully. In TrojAI Leaderboards, there are some better results.
- **Methodological Clarity**: Some steps in the methodology, particularly the heuristic search for Trojan triggers, could be explained in more detail to enhance reproducibility.
- **Parameter Sensitivity Testing**:The study lacks an analysis of how sensitive the Memorization Score is to various parameters, such as threshold values used in Trojan detection or the number of suffix proposals in training data extraction.

---

### Official Review · Reviewer_1ypk · 2024-10-09
**The paper could benefit from a clearer threat model, more detailed explanations, and a clearer demonstration of the performance.**

**Rating:** 4
**Confidence:** 4

**Review:**

This paper introduces a novel technique that utilizes Mutual Information (MI) to measure memorization. Despite its merits, there are several aspects that require enhancement.

Firstly, it seems that a clear threat model is missing for the Trojan detection. Additionally, the description in the implementation details section makes it somewhat challenging to understand how MS is applied for detection. It might be helpful if the paper provided more details or perhaps included some figures to clarify this process.

Regarding the training data extraction, based on Table 2, the proposed method does not seem to significantly outperform the baselines. For instance, comparing the values 50.3 with 49.6 and 77.8 with 77.5 shows only marginal differences. Would it be possible to explore combining MS with zlib or high-conf for improved results? Moreover, testing the methods on different datasets could provide further insights.